# Baseline Cytokine Profile Identifies a Favorable Outcome in a Subgroup of Colorectal Cancer Patients Treated with Regorafenib

**DOI:** 10.3390/vaccines11020335

**Published:** 2023-02-02

**Authors:** Andrea Abbona, Vincenzo Ricci, Matteo Paccagnella, Cristina Granetto, Fiorella Ruatta, Carolina Cauchi, Danilo Galizia, Michele Ghidini, Nerina Denaro, Marco Carlo Merlano, Ornella Garrone

**Affiliations:** 1Fondazione Arco Cuneo, 12100 Cuneo, Italy; 2Medical Oncology Unit, AORN “San Pio”, 82100 Benevento, Italy; 3Azienda Ospedaliera S. Croce e Carle, 12100 Cuneo, Italy; 4Department of Medical Oncology, Fondazione IRCCS Ca’ Granda Ospedale Maggiore Policlinico, 20122 Milano, Italy; 5Candiolo Cancer Institute, FPO-IRCCS, 10060 Candiolo, Italy

**Keywords:** cytokinome, regorafenib, mCRC, cytokine profile, PCA

## Abstract

Metastatic colorectal cancer is frequently associated with poor clinical conditions that may limit therapeutic options. Regorafenib is a small molecule approved for the treatment of metastatic colorectal cancer, but it is hampered by significative toxicities. Moreover, only a relatively limited number of patients benefit from the treatment. Therefore, the identification of reliable markers for response is an unmet need. Eighteen cytokines, selected based on their prevalent Th1 or Th2 effects, were collected. Peripheral blood samples were gathered at baseline in 25 metastatic colorectal cancer patients treated with regorafenib. Data extracted have been linked to progression-free survival. ROC identified the best cytokines associated with outcome. The relative value of the selected cytokines was determined by PCA. Data analysis identified 8 cytokines (TGF-β, TNF-α, CCL-2, IL-6, IL-8, IL-10, IL-13 and IL-21), used to create a signature (TGF-β, TNF-α high; CCL-2, IL-6, IL-8, IL-10, IL-13 and IL-21 low) corresponding to patients with a significantly longer progression-free survival. This report suggests that the analysis of multiple cytokines might identify a cytokine signature related to a patient’s outcome that is able to recognize patients who will benefit from treatment. If confirmed, future studies, also based on different drugs, using this approach and including larger patient populations, might identify a signature allowing the a priori identification of patients to be treated.

## 1. Introduction

Colorectal cancer (CRC) is the third most common cancer and the second leading cause of cancer-related deaths in Western countries [1]. About 20% of CRC patients have metastatic disease at diagnosis and 50% of patients with stage III show disease recurrence after front-line therapy. Frequently, metastatic CRC (mCRC) patients are debilitated [2] and it may limit the therapeutic options and make treatment complex due to the potential treatment-related toxicities [3]. All these aspects explain why mCRC is historically associated with a dismal prognosis, with 5 to 8% overall survival (OS) at 5 years.

Regorafenib is an oral multi-kinase inhibitor approved for the therapy of previously treated mCRC [4]. Its main effect is the suppression of angiogenesis and remodeling of the tumor microenvironment (TME), favoring antitumor immunity via blockade of vascular endothelial growth factor receptors (VEGFRs) (accelerates maturation of dendritic cells, improves cytotoxic T lymphocyte trafficking and cytotoxic function) and colony stimulating factor 1 receptor (CSF-1R) (reduced TAM recruitment and differentiation toward the M1 phenotype), among other positive immune effects (4).

Pivotal clinical trials have shown that regorafenib could significantly increase OS and progression-free survival (PFS) compared with placebo [5,6,7]. Currently, regorafenib is recommended as the third or following line of therapy [8]. 

At the standard approved dose, regorafenib is frequently associated with adverse effects (AEs) [9,10] and it limits its use.

Since only a fraction of mCRC patients show clinical benefits with regorafenib, there is an unmet need for biomarkers able to identify responder patients, avoiding the risk of toxicity to the remaining population [11]. 

The interaction between the tumor and immune cells is modulated by the TME and is largely driven by small proteins (interleukines, growth factors, chemokines, for brevity all them called cytokines) detectable in peripheral blood [12]. Their analysis offers a tool for associating specific cytokine profiles with the likelihood of obtaining a benefit from treatment. 

Our group has already investigated the relationship of single cytokines with clinical outcome finding TGF-β and TNF-α both associated with outcome [13]. However, single cytokine analysis is a poor method for identifying effective markers because their final effect is frequently context dependent [14].

Therefore, a study of panels of cytokines selected among those potentially more involved in the determination of TME should be favored [12]. 

For this reason, we decided to evaluate a panel of cytokines in patients treated with regorafenib, a drug known to influence TME, using statistical methods capable of highlighting their mutual influence. Basically, this analysis points out the effect of context rather than the value of individual cytokines.

The aim of the study is to identify a cytokine profile associated with longer PFS, which represents a simple measure of drug effect [15].

## 2. Materials and Methods

### 2.1. Study Design

This is an exploratory retrospective single-center study conducted at the Department of Clinical Oncology, Ospedale S. Croce e Carle teaching hospital. 

### 2.2. Patients

Patients considered for the study suffered from histologically confirmed mCRC already treated with standard regimens of chemotherapy. All patients were required to be eligible for treatment with regorafenib (Stivarga^®^, Bayer, Leverkusen, Germany), according to EMA approval. All enrolled patients signed an informed consent for the storage and analysis of their biological material and the study was approved by the local ethical committee (prot n° 24347; 7 August 2015).

### 2.3. Blood Sample Collection

Blood samples were collected into EDTA vacutainer tubes at baseline immediately before the first administration of regorafenib. Plasma samples were obtained through the centrifugation step and stored in aliquots at −80 °C until use.

### 2.4. Analysis Methods

#### 2.4.1. Plasma Levels of 18 Cytokines

TGF-β, TNF-α, VEGF, INF-γ, IL-2, IL-4, IL-5, IL-6, IL-8, IL-10, IL-12, IL-13, IL-15, CCL-2, CCL4, CCL-22, and CXCL-10 were evaluated with the Ella Simple Plex system (ProteinSimple™, San Jose, CA, USA) according to the manufacturer’s instructions. Briefly, a twofold dilution of each plasma sample was spun for 15 min at 1000× *g* and added to the Simple Plex cartridge. The cartridge was then inserted into the reactor and run for 90 min at RT. TGF-β, was previously activated (1 N HCl, and then neutralized with 1.2 N NaOH/0.5 M HEPES) to a final dilution with a volume ratio of 1:15. The cartridge was inserted into the Ella reactor and run for 90 min. The concentrations were expressed in pg/mL.

#### 2.4.2. IL-21

IL-21 was assessed with the ELISA method (R & D System, Minneapolis, MN, USA). The Il-21 reaction, after incubations, was stopped and colorimetric detection was carried out with a spectrophotometer (Multiskan Ascent, Thermo Fisher Scientific, Cambridge MA, USA) set at 450 nm with corrections at 570 nm. The measured optical densities were expressed as pg/mL.

### 2.5. Statistical Analysis

We retrospectively grouped patients into 2 groups based on the median value of PFS (above or below the median value).

Differences in the median cytokine values were analyzed using a non-parametric Mann–Whitney U test. In order to find the optimal cut-off point of our variables, receiver operating characteristic curve (ROC) analysis was performed, also considering variables with *p*-values below 0.2. The cut-off was defined as the point on the ROC curve with the largest average sensitivity and specificity. 

Principal component analysis (PCA) and hierarchical clustering on principal components (HCPC) were performed to group our populations into clusters, using normalized variables with the z-score method. To realize these clusters, the “elbow method” was employed to cut the hierarchical tree. 

PFS and OS were evaluated using the Kaplan–Meyer method, and the relative hazard ratio (HR) was analyzed with the Cox model.

Response was assessed every 8 weeks and classified according to RECIST version 1.1. Clinical benefit (CB) was defined as the sum of all complete responses (CR), partial responses (RP) and stable diseases (SD) lasting at least 6 months.

PFS was defined as the time elapsed between the start of regorafenib and the diagnosis of progression of disease or death from any cause, whichever occurred first or at the date of the last follow-up for censored patients. 

OS was defined as the time elapsed between the start of regorafenib and death from any cause or the date of the last follow-up for censored patients.

The Mann–Whitney U test was performed with GraphPad v.5. Kaplan–Meyer analysis was performed with STATA MP13 and the Cox model was performed with SPSS V.24. PCA and HCPC were used to identify different clusters of patients based on specific cytokine profiles and were computed with R v.3.5.3 by the FactoMiner R package. In all tests, a *p* value equal to or lower than 0.05 was regarded as significant. Bonferroni’s correction was applied to the multiplicity test [16]. If not specified, a *p*-value was considered NS (not significant).

## 3. Results

### 3.1. Patient Population

Twenty-five patients accrued between June 2015 and September 2020; the median age was 65 years. Fifteen patients (60%) had a primary site in the colon (10 patients left colon and 5 right colon) and 10 patients (40%) in the rectum. Twenty patients (80%) harbored a RAS mutation, and five (20%) were RAS wild type. The main patients’ characteristics are reported in Table 1.

All the patients received at least 2 previous lines of treatment for metastatic disease, and 10 patients received 4 or more lines. Previous treatments according to type of treatment, line of treatment and patient’s clusters are reported in Table 2. 

The most common site of metastatic deposit was the lung (11/13 patients in cluster 1, 6/11 patients in cluster 2 and 1/1 patient in cluster 3). The liver was involved in 8/13 patients in cluster 1, 8/11 patients in cluster 2 and 1/1 patient in cluster 3. Three patients, all in cluster 2, had only unresectable liver metastases and 2 patients, all in cluster 1, had only multiple unresectable lung metastases. One patient (cluster 2) had only peritoneal involvement. In all the remaining patients, multiple metastatic sites were observed. The metastatic spread is described in Appendix A. 

### 3.2. Treatment Effect

One patient achieved CR; clinical benefit was recorded in 5 patients (20%). Twenty patients experienced PD in the first evaluation.

The median PFS was 2.9 months (95% C.I. 2.2–3.5), and the median OS was 9.1 months (95% C.I. 6.5–11.7). 

### 3.3. Correlation between Baseline Cytokine Levels and PFS

Cytokine levels were assessed in all accrued patients.

Plasma levels of TGF-β (*p* = 0.01), IL-6 (*p* = 0.01), IL-8 (*p* = 0.04), IL-10 (*p* = 0.03) and TNF-α (*p* = 0.04) were higher in patients below the PFS median value. CCL-2 (*p* = 0.001) and IL-21 (*p* = 0.001) plasma levels were higher in patients with PFS above the median (Figure 1).

### 3.4. Cluster Analysis

Using ROC analysis, we identified 8 cytokines with a prognostic value. Seven cytokines showed a significant specificity for patients with PFS below the median compared to patients with PFS above the median (TGF-β, TNF-α, CCL-2, IL-6, IL-8, IL-10 and IL-21). IL-13, with a *p*-value of 0.109, was also selected (Appendix A).

PCA was realized based on the identified cytokines (TGF-β, TNF-α, CCL2, IL-6, IL-8, IL-10, IL-13, and IL-21), and a graph of the cytokine vector distribution was divided into four quadrants (Figure 2).

Quadrant 1 was generated with IL-21 and CCL-2 vectors; quadrant 2 with IL-13, IL-8 and TNF-α vectors; quadrant 3 with the remaining cytokines (IL-6, IL-10 and TGF-β); no cytokines contributed in quadrant 4. Then, a factor map was developed using HCPC analysis. All patients were grouped into 3 clusters: cluster 1 (13 patients), cluster 2 (11 patients) and cluster 3 (1 patient) (Figure 3).

### 3.5. Kaplan–Meier Analysis

Considering that cluster 3 is made up of only one patient, analyses were focused between cluster 1 and cluster 2. 

A significantly higher median PFS was observed in cluster 1 (5.2 months, 95% CI: 4.0–6.4, vs. 2.4 months, 95% CI: 2.3–2.5, *p* < 0.001) (Figure 4). 

A multivariate Cox analysis was performed using clusters (used as dichotomous variables), sex, primary site (left colon, right colon or rectum), RAS mutational status and number of prior therapies as covariates. The belonging in cluster 1 was the only independent factor predicting good PFS, with an HR of 0.110 (95% C.I. 0.030–0.399) (Table 3). 

## 4. Discussion

This study allowed the identification of a profile made up of 8 cytokines, able to discriminate, in our limited population, patients with the best PFS.

The profile has been identified using PCA and is based on high levels of CCL-2 and IL-21 and low levels of TGF-β, IL-10, IL-6, IL-8, TNF-α and IL-13. 

HCPC analysis identified each single patient harboring the good profile and placed them in a cluster characterized by significantly better PFS (cluster 1).

The value of this approach is that it overcomes the context-dependent effect of most cytokines, highlighting their mutual influence.

Considering, for instance, the 8 cytokines of the identified panel, 7 of them can induce conflicting effects.

CCL-2 has been associated with a poor prognosis in breast cancer [17] and may support the formation of metastatic niches through the recruitment of monocytes [18]. However, it also recruits γδ T lymphocytes, favoring immune surveillance [19].

IL-21 favors maturation and enhances the cytotoxicity of CD8+ cells and NK cells, while suppressing the induction of Tregs [20,21,22]. However, it may also negatively impact γδ T cell anti-tumor effects [23].

TGF-β is considered among the most immunosuppressive cytokines due to its major role, for instance, in extracellular matrix remodeling, contribution to neo-angiogenesis and epithelial-mesenchymal transition (EMT) [24]. However, TGF-β has important tumor-suppressor effects, in particular in the initial phase of cancer development or in tumors with preserved TGF-β signaling, by acting negatively on cell proliferation and positively on apoptosis [25].

IL-10 promotes immune suppression in TME [26,27], but surprisingly, the reduction predicts poor outcomes in lung and colon cancer [28,29].

IL-6 is a negative prognostic factor in many solid tumors, including colorectal cancer [30]. However, IL-6 shows an important tumor-suppressor effect based on the activation, expansion and survival of effector lymphocytes [31].

IL-8, among other pro-tumor effects, contributes to EMT and promotes trafficking of myeloid-derived suppressor cells (MDSC) and neutrophils [32] toward the tumor bed. Conversely, Doll et al. reported that IL-8 was not related to tumor progression or poor prognosis in colorectal cancer [33].

TNF-α promotes apoptosis in immune cells and favors tumor dissemination [34]. On the other hand, it contributes to the M2 (pro-tumor)–M1 (tumor suppressor) conversion of tumor-associated macrophages and to the destruction of tumor vasculature [35].

Among the cytokines belonging to the panel identified, only IL-13 seems to harbor exclusively pro-tumor effects. It is involved in EMT, acts as a growth factor in pancreatic cancer and other solid tumors, is an important regulator of M2 macrophages damping immune surveillance against metastasis [36], and favors lymph node dissemination [37]. 

However, due to the conflicting effects of most of them, the final role of cytokines is believed to be context dependent; therefore, the concurrent analyses of multiple cytokines, instead of a single one, using adequate statistical methods, may contribute to a more accurate assessment of a patient’s immunological status and prognosis [14,38].

Chen et al. selected 17 cytokines correlated to OS at univariate analysis into a cytokine score [39]. The authors assigned a weighted score to the cytokines based on their respective HR. The cut-off value for the score was then measured by ROC to transform the variable into a dichotomous high and low value. Sensitivity and specificity in predicting OS were 0.833 and 0.737 values, respectively. 

Unfortunately, this study did not evaluate the scores against important covariates, such as tumor or patient characteristics, in a multivariate model. However, the cumulative analysis of the 17 cytokines was better than that of all single cytokines in terms of prognostic accuracy. Gunawardene et al. in 2018, performed a systematic review focusing on the prognostic value of circulating cytokines in colorectal cancer. They concluded that evaluating multiple cytokines is relatively ineffective for identifying novel biomarkers, albeit the levels of multiple cytokines combined into a composite score might be promising [40]. These conclusions highlight the need to apply adequate statistical methods to this issue. 

Several studies have tried to employ PCA, a method able to reduce the dimensionality of multi-variable data while enabling an unbiased data-driven approach, to investigate the efficacy of treatments and/or understand an unbalanced immune system that drives early progression or death. For instance, Tuong et al. used PCA to assess a cytokine profile related to the disease stage in squamous cell carcinoma and precancerous lesions of the skin [41]. Nistor et al. analyzed, with the same method, a cytokine network in a randomized phase II study in metastatic melanoma patients treated with dendritic cell vaccines or tumor cell vaccines to compare the immune responses of each treatment [42]. Ellsworth et al. used PCA to identify changes in cytokine profiles in non-small cell lung cancer patients treated with definitive radiotherapy [43]. 

Our group has used PCA to identify a cytokinome signature able to discriminate different prognostic groups among end-stage patients affected by different solid tumors [44] and in breast cancer patients to identify a signature potentially able to select patients for treatment beyond progression [45]. However, to the best of our knowledge, this is the first study on colorectal cancer using PCA to evaluate a cytokine profile in patients with metastatic disease treated with standard therapy. 

## 5. Conclusions

Our report is hampered by a key limitation: the results cannot be used routinely in clinical practice, as they are and must be regarded as exploratory. 

Indeed, PCA represents a dynamic method and the addition of new data may change the relationship among the analyzed cytokines, leading to different results. 

It means that a much larger number of patients should be analyzed to reduce the variability and to obtain more stable results.

One other important limitation is that we cannot weigh the importance of the treatment. It is highly probable that different drugs or combinations may change the predictive role of the analyzed cytokines. To assess this point, we are testing the same 18 cytokines selected for the present experience in ongoing projects conducted on patients treated with different drugs and with different solid tumors. Notwithstanding these limitations, the study underlines that PCA may allow the identification of a cytokine signature related to a patient’s outcome, highlighting the contextual rather than the context-dependent effect of each cytokine.

The next step, related to the results reported, is to build up a network of centers able to enroll in and analyze a large number of homogeneously treated colon cancer patients. 

Our ambitious aim is to identify a cytokine signature able to prospectively drive the treatment choice. It is important to underline that the technology used in our study is relatively low in cost and easily expanded to most hospitals other than research institutions.

## Figures and Tables

**Figure 1 vaccines-11-00335-f001:**
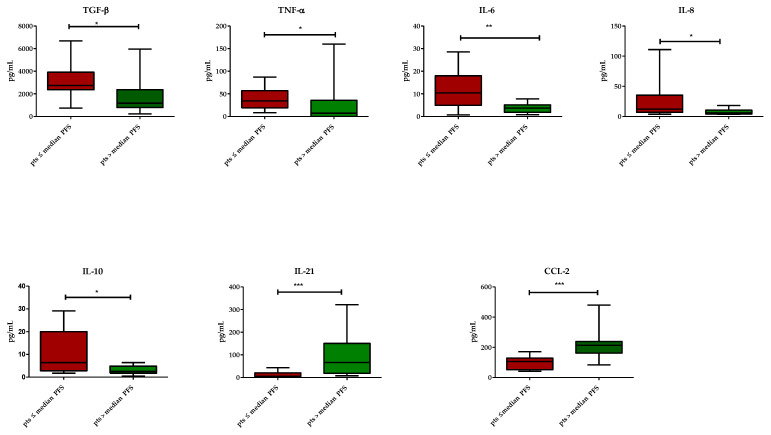
Distributions of cytokines: red bars represent patients with PFS ≤ 2.9 months and green bars represent patients with PFS > 2.9 months. Only cytokines with any statistical significance were shown. Data are expressed as medians with ranges. *** *p* < 0.001, ** *p* < 0.01, * *p* < 0.05. CCL, C-C motif ligand; IL, interleukin.

**Figure 2 vaccines-11-00335-f002:**
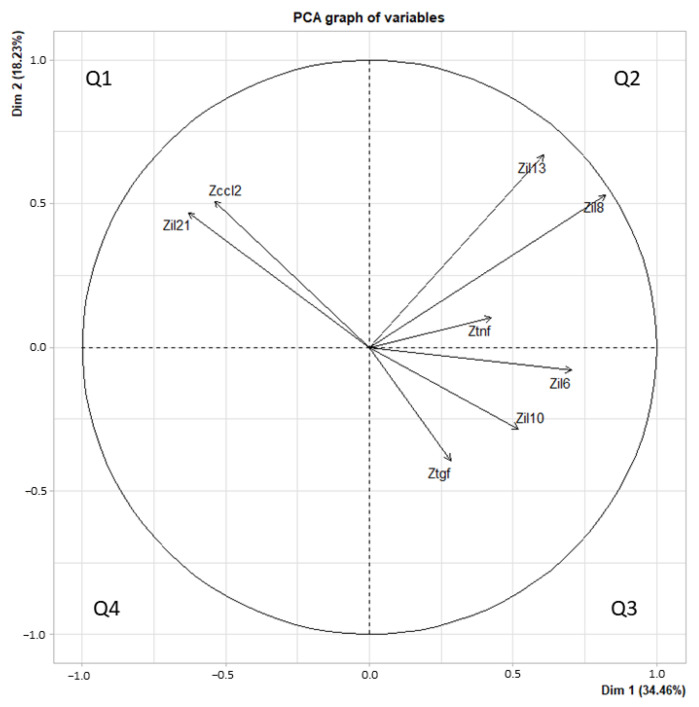
PCA graph of variables for each quadrant. Quadrant 1 (Q1) was represented by CCL-2 and IL-21; quadrant 2 (Q2) by IL-13, IL-8 and TNF-α; quadrant 3 (Q3) by IL10, IL-6 and TGF-β. The *x*-axis plotted principal component 1 (Dim 1), and the *y*-axis plotted principal component 2 (Dim 2). Variance explained was represented as a percentage.

**Figure 3 vaccines-11-00335-f003:**
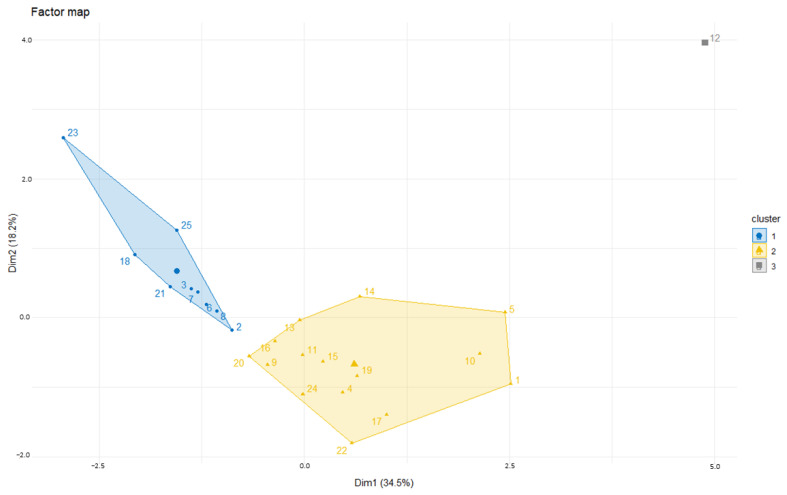
HCPC analysis for all 25 patients. In the x- and y-axes, the two principal components are plotted (Dim 1 and Dim 2). We clustered all patients in 3 clusters: cluster 1 (blue, 13 patients), cluster 2 (gold, 11 patients) and cluster 3 (gray, 1 patient).

**Figure 4 vaccines-11-00335-f004:**
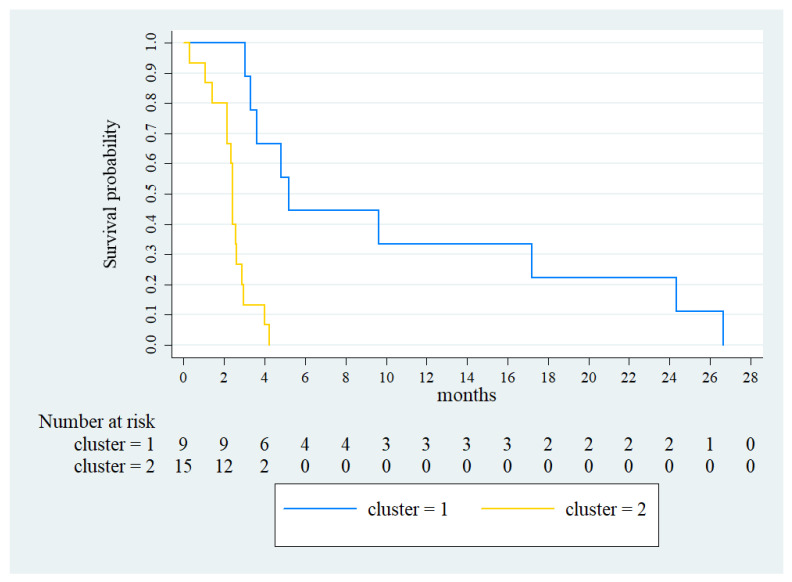
Kaplan-Meier for the PFS of cluster 1 (blue line) and cluster 2 (gold line).

**Table 1 vaccines-11-00335-t001:** Patient’s characteristics.

Characteristics	Number 25
Age (median, range)	65 (48–80)
ECOG PS (median, range)	0 (0–1)
Sex	
Male	14 (56%)
Female	11 (44 %)
Primary Tumor Site	rectum 10 (40%)
	colon 15 (60%)
Mutational RAS status	
Mutated	20 (80%)
Wild type	5 (20%)
Previous anticancer therapies in individual patients	
2	4 (16%)
3	11 (44 %)
4	8 (32 %)
≥5	2 (8 %)
Median n (range)	3 (2–7)

Legend: ECOG PS, Eastern Cooperative Oncology Group performance status.

**Table 2 vaccines-11-00335-t002:** Prior treatments.

Treatment Line	Treatment			Patients (# by Cluster)			Patients
	Cluster 1	Cluster 2	Cluster 3	Cluster 1	Cluster 2	Cluster 3	Total
1	Folfiri ^1^ + beva ^2^ Xelox ^3^ + beva ^2^ Folfox ^4^ Folfoxiri ^5^ + beva ^2^ Folfox ^4^ + beva ^2^ Xelox ^3^ Cape ^6^ + beva ^2^	Xelox ^3^ Folfoxiri ^5^ + beva ^2^ Folfox ^4^ Xelox ^3^ + RT Folfox ^4^ + beva ^2^ Folfoxiri ^5^	Xelox ^3^ + beva ^2^	13	11	1	25
2	Folfiri ^1^ (re-) ^9^ Xelox ^3^ Cape ^6^ Folfiri ^1^+ beva ^2^ Cape ^6^ + beva ^2^ Fu ^10^ + beva ^2^ Xeliri ^7^ Folfiri ^1^	Folfiri ^1^ Cape ^6^ + beva ^2^ Folfiri ^1^ + beva ^2^ Xelox ^3^ + beva ^2^ Folfiri ^1^ + afli ^12^	Folfiri ^1^ + beva ^2^	13	11	1	25
3	Folfox ^4^ FU ^10^ + beva ^2^ Xelox ^3^ Cet ^8^ + Irino ^11^ Folfox ^4^ + beva ^2^ (re-) ^9^ Folfiri ^1^ + beva ^2^ Irino ^11^ + beva ^2^ Lons. ^14^	Folfox ^4^ Folfiri ^1^ + beva ^2^ Folfox ^4^ (re-) ^9^ Cape ^6^ Lons. ^14^	Lons. ^14^	12	8	1	21
4	Folfiri ^1^ + Cet ^8^ Folfox ^4^ Folfox ^4^ + afli ^12^ Pani ^13^ Lons. ^14^ Folfox ^4^ + beva ^2^	Pani ^13^ Xelox^3^ + beva ^2^ Cape ^6^		7	3	-	10
5	Folfox ^4^ (re-) ^9^	Irino ^11^ + beva ^2^		2	1	-	3
6	Irino ^11^ + cet ^8^			1	-	-	1
7	Lons. ^14^			1	-	-	1

Legend: ^1^ Folinic acid + fluoruracil + irinotecan; ^2^ bevacizumab; ^3^ capecitabine + oxaliplatin; ^4^ fluorouracil + folinic acid + oxaliplatin; ^5^ folinic acid + fluorouracil + oxaliplatin + irinotecan; ^6^ capecitabine; ^7^ capecitabine + irinotecan; ^8^ cetuximab; ^9^ rechallenge; ^10^ fluorouracil; ^11^ irinotecan; ^12^ aflibercept; ^13^ panitumumab; ^14^ trifluridine-tipiracil.

**Table 3 vaccines-11-00335-t003:** Multivariate Cox analysis for PFS.

Variable	HR	95% C.I.	*p* Value
Cluster 1 (ref Cluster 3)	0.110	0.030–0.399	0.001
Sex (ref male)	0.867	0.428–2.737	0.867
Primary site (ref rectum)			0.555
Right Colon	0.528	0.127–2.195	0.380
Left Colon	1.481	0.390–5.630	0.564
RAS status (ref wild type)	0.670	0.173–2.595	0.562
N° line of therapies (ref > 3 lines)	2.067	0.606–7.056	0.246

Legend: HR hazard ratio, C.I. confidence interval.

## Data Availability

Data supporting reported results can be found at the ARCO foundation laboratory in the Santa Croce e Carle Teaching Hospital (Cuneo, Italy).

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
