# Peer review of "Baseline Cytokine Profile Identifies a Favorable Outcome in a Subgroup of Colorectal Cancer Patients Treated with Regorafenib"

_vaccines, 2023, doi:10.3390/vaccines11020335_

Round 1

Reviewer 1 Report

The authors used multiplex cytokine level measurement and PCA to determine a cytokine pattern in such patients with metastatic CRC who ultimately failed multiple previous treatment regimens, were treated with regorafenib, and responded well to treatment.
The authors describe the study as an exploratory, retrospective, single-center study. 
Although their ambition and results encourage further research, several aspects were not taken into account, which makes the true value of their results questionable. 
The number of patients included in the study is very small. A total of 25 patients were studied, who formed an additional 3 subgroups (right and left CRC, rectal cc.). In any case, more patients would need to be included in order to give credibility to the true clinical value of the results obtained by the complex statistical method. 
The majority of patients had received three or more previous treatments. It is not known exactly which treatments these were. This would be important to know, among other reasons, because not only chemotherapy but also biological treatments can have a major impact on the cytokine patterns observed in the circulation of patients with tumors. I would consider the results much more credible if the study had included patients who were enrolled in a therapy naïve state and had their cytokine profiles examined before and after each treatment cycle. This would give a truer picture of the distorting effect of treatments on cytokine patterns. 
It would also have been good to know which metastases mCRC patients have: for example, metastatic involvement of the liver is associated with alterations in immunological mechanisms different from those of the lung. 
Not to mention that the histological, genetic, and epigenetic pattern of colorectal cancer also has a significant impact on the tumor-microenvironment and tumor-host immune system interactions. 
It would also have been good to include a control group: how would the cytokine patterns of mCRC patients have changed with other therapies? 
Subsection 3.1 is part of the Methodology, not the Results. 
Overall, the results are interesting, but despite the complex statistical analyses, they cannot be called clinically certain results. 
I do not consider the manuscript in its present form suitable for publication. 

Author Response

Rewiever n. 1

Comment: “The number of patients in the study is very small…

Answer: We totally agree. The study indeed is only exploratory and we are aware that it only may generate a working hypothesis. We hope to be able to plan a larger prospective study in the near future.

Comment: “the majority of patients had received three or more previous treatments. It is not known exactly which treatments these were.”

Answer:  The Reviewer points out a serious shortcoming of the manuscript. We have added details regarding prior treatments in the text (3.1 Patients population) and also in a dedicated table (tab. N. …)

Comment: “I would consider the results much more credible if the study had included patients who were enrolled in a therapy naïve state and had cytokine profiles examined before and after each treatment cycle.”

Answer: in our study the drug considered is regorafenib. This drug is not approved for use in naïve patients. We could have used a different drug, but decided on regorafenib for its effects on the immune microenvironment.

The use of multiple determinations of circulating cytokines, before and after each course of therapy, is an excellent way to study the effect of treatment on the circulating elements of the TME.

Our goal is more modest but also easier to transfer to the clinic: looking for a cytokine profile able to identify a subpopulation of patients suitable to achieve a benefit from regorafenib therapy.

Comment: “It would also have been good to know which metastases mCRC patients have: for example, metastatic involvement of the liver is associated with alterations in immunological mechanisms different from those of the lung.”

Answer: This again is absolutely true. Therefore, we have added information about metastatic sites as suggested by the Reviewer both in the text (3.1 Patients population) and in a dedicated table (Table S1). However, we like to underline that the measurement of cytokines from circulating blood offers only an average of the production of cytokines from the different metastatic tumor microenvironments, and cannot distinguish a specific one. Moreover, as already demonstrated in humans (for example by Alejandro Jimenez-Sanchez, Cells 2017), heterogeneous immune microenvironments exist among metastases in the same patient even if they grow in the same organ (including liver).

Comment: ‘It would also been good to include a control group…’

Answer: It is again a correct comment. Indeed, a control group including patients with different therapies may help to understand the role of specific drug/combination. However, the effect of different therapies could also be evaluated in future, because a similar approach would not be a sort of “randomized study” but only a not formal comparison. 

Comment: ‘Subsection 3.1 is part of the methodology, not the results’

Answer: We did not understand this comment. Subsection 3.1 does not describe the characteristics of the patients to be accrued, but the patient population really enrolled into the study and, as such, it represents a “result”.

In addition reviewer n.1 suggests to improve "introduction", citations, study design, description of methods and conclusions. We have provided additional information in the introduction, we have added two new references and we have described more accurately the methods (all the changes typewrote in red). These changes in our opinion facilitate the understanding the research design and better support the results.

Reviewer 2 Report

In this paper, the authors have aimed to identify the reliable marker for response onto regorafenib. The authors identified the signature using 8 cytokines (TGF-β, TNF-α, CCL-2, IL-6, IL-8, IL-10, IL-13 and IL-21) corresponding to patients with a significant longer progression free survival. Although, this is a reasonably well written manuscript, following concerns need to be addressed by the authors prior to the publication.

In this study, all subjects were already treated with standard regimens of chemotherapy prior to regorafenib administration and serum cytokine levels were compared only before the regorafenib administration. Therefore, the difference in cytokine profile between Cluster1 and Cluster2 may only reflect the status of patients before administration of regorafenib.

If the authors aim to identify the cytokine profile which represents the effect of regorafenib administration as described in title, the cytokine profile must be evaluated before and after administration of regorafenib. Title is misleading.

And also, the Authors need to confirm whether the description “ref Cluster 3” in Table 2. is not a mistake of “Cluster 2”.

Author Response

Reviewer N. 2

Comment: ‘…Therefore the difference in cytokine profile between cluster 1 and cluster 2 may only reflect the status of patients before administration of regorafenib. … Title is misleading.’

Answer: The comment is correct. Unfortunately, the title was misleading. We have changed the title and we hope that it now reflects more clearly our aim. Our purpose was not the identification of the changes induced by regorafenib on circulating cytokines, but to identify a basaline cytokine profile able to identify a subpopulation of patients suitable to achieve a benefit from regorafenib therapy. 

Tab. 2 (Table 3 in the edited manuscript) has been corrected: we thank the Reviewer.

In addition the reviewer n.2 suggest to improve the "introduction", research design and conclusions. We have already answered to reviewer n.1 to these points.

Reviewer 3 Report

In this paper entitled “Cytokine profile identifies a favourable outcome in colorectal cancer patients treated with regorafenib” aim to identify a cytokine profile associated with longer PFS which represents a simple measure of drug effect. This is important for future clinics and provide important check points about treatment.

The paper has a modest interest, since the authors found similar results previously. What is the added value with this increased panel? The authors could increase the number of patents. The discussion is merely descriptive and not properly discussion.  

Author Response

Reviewer N. 3

Comment: ‘The study has a modest interest, since the authors found similar results previously. What is the added value with this increased panel? ...’

Answer: We appreciate this comment because it has revealed a confounding aspect of the manuscript in its previous form. We have modified or added sentences hoping to clarify the added value of this report, which is the use of statistical methods able to ‘overcome the context-dependent effect of most cytokines highlighting their mutual influence.’ (Discussion, lines 7 – 8 from the top).

We have added data from the literature, focusing on the limitation of single cytokine analysis for the identification of new markers, but also the unsatisfactory results achieved by the analysis of multiple cytokines, if adequate statistical methods are not used.

We also highlighted that, at the best of our knowledge, our study is the first one using Principal Component Analysis (PCA) to identify a subpopulation of metastatic colorectal cancer patients through a specific cytokine profile able to predict a good outcome following a standard treatment. 

Comment: ‘The discussion was merely descriptive and not properly a discussion.’

Answer: Discussion section has been improved to answer the previous comment and we hope that in its present form it may join the Reviewer’s expectation.

In addition reviewer n.3 suggest to improve research design, methods, results and conclusion. Again, we believe that the changes to reviewer n.1 should have answered these points.

Round 2

Reviewer 1 Report

The authors have corrected and completed their manuscript on very important aspects. As a result, the manuscript has undergone significant improvements in quality and presentation.
The corrected version addresses all previously unanswered questions and, where no data are available, provides a clear and acceptable explanation.
I recommend that the manuscript be accepted for publication.

Reviewer 3 Report

The authors have performed the reviewer suggested corrections.